# MicroRNAs profiling in malaria and arbovirus coinfection: A systematic review protocol

Andrillene Laure Deutou Wondeu [1]*, Alex Durant Nka[2], Aude Christelle Ka'e[2], Ezechiel Ngoufack Jagni Semengue[2], Sylvanie Masso[1], Pascal Fisemou[1], Adegoke Taiwo Mobolaji[1], Zeekah Marymag[1], Calvino Fomboh Tah[1], Rhoda Nsen Bughe[1], Akindeh Mbu Nji[1], Andrea Galgani[3], Stefano Pirrò[3], Vittorio Colizzi[2,3], Wilfred Fon Mbacham[1]

1 The Laboratory for Public Health Research Biotechnologies, MARCAD Plus Program, The Biotechnology Center, University of Yaoundé 1, Yaoundé, Cameroon, 2 Chantal Biya International Reference Centre for Research on HIV/AIDS Prevention and Management, Yaoundé, Cameroon, 3 Department of Biology and Interdepartmental Center for Comparative Medicine, University of Rome Tor Vergata, Rome, Italy

* andrillene.1@gmail.com

## Abstract

### Background

Coinfections between malaria and arboviruses such as Dengue virus (DENV), chikungunya Virus (CHIKV), and Zika virus (ZIKV) represent a significant clinical and public health challenge. The molecular pathogenesis of these coinfections is complex and poorly understood. MicroRNAs (miRNAs) are key regulators of gene expression and play a crucial role in the host response to infection. A comprehensive profile of miRNAs implicated in malaria-arbovirus coinfection could provide novel insights into disease mechanisms and reveal new targets for improved management and therapeutic strategies.

### Methods

The review will be conducted in accordance with Preferred Reporting Items for Systematic Reviews and Meta-Analyses (PRISMA) guidelines. A comprehensive search strategy will be executed in electronic databases (PubMed/MEDLINE, Scopus, Web of Science, Embase, CENTRAL, CINAHL) from January 2000 onwards, with no language restrictions. Grey literature sources will also be searched. Included study designs will comprise observational studies (cohort, case-control, cross-sectional) and clinical trials reporting primary data on miRNA expression. Two independent reviewers will screen records, extract data, and assess the risk of bias using appropriate tools (ROBINS-I, ROB-2, NIH tools). Data on study characteristics, patient demographics, confirmation of coinfection, miRNA profiling techniques, and differentially expressed miRNAs will be extracted. Given anticipated heterogeneity, a

provided the original author and source are credited.

**Data availability statement:** No datasets were generated or analyzed during the current study. All relevant data from this study will be made available upon study completion.

**Funding:** The author(s) received no specific funding for this work.

**Competing interests:** The authors have declared that no competing interests exist.

**Abbreviations:** AGO, Argonaute; CHIKV, Chikungunya virus; DENV, Dengue virus; MARCAD, Malaria Research Capacity Development Consortium; MeSH, Medical Subject Headings; MiRISC, miRNA-induced silencing complex; miRNAs, MicroRNAs; mRNAs, messenger ribonucleic acid; PICO, patients, intervention, comparison, outcome; pre-miRNAs, precursor microRNAs; PRISMA-P, Preferred Reporting Items for Systematic Reviews and Meta-Analysis; ZIKV, Zika virus; 3′-UTR, 3′-untranslated region

narrative synthesis will be performed. The strength of the evidence will be assessed using the GRADE approach.

## Conclusion

This systematic review will provide comprehensive evidence on miRNA expression patterns associated with malaria-arbovirus coinfections. The findings will advance understanding of coinfection pathogenesis and identify potential biomarkers for improved diagnostics and therapeutics. It will also pioneer an exploration of cross-kingdom gene regulation by plant-derived miRNAs, establishing a foundation for future functional validation studies. Additionally, the miRNA profiles identified may provide foundations for developing novel therapeutic approaches, including nanoparticle-based delivery systems.

## Systematic review registration

PROSPERO registration number: PROSPERO CRD42024600379.

## Introduction

MicroRNAs are a class of endogenous non-coding ribonucleic acids, consisting of around 20 nucleotides. Their role is to regulate the post-transcriptional expression of genes by either inhibiting or activating the translation of targeted proteins, depending on gene complementarity [1]. The canonical pathway of miRNA biogenesis begins in the nucleus of the cell and ends in the cytoplasm. Through biomolecular and enzymatic processes, they are transcribed from DNA sequences into primary miRNAs and transformed into precursor miRNAs (pre-miRNAs) in the nucleus. These pre-miRNAs are exported to the cytoplasm, where the mature miRNA duplex biosynthesis is finalized. Finally, the 5p or 3p strands of the mature miRNA duplex are loaded into the Argonaute (AGO) family of proteins to form a miRNA-induced silencing complex (miRISC) [2,3].

Mechanisms through which miRNAs regulate gene expression involve the interaction of their seed sequences (the first 2–7 nucleotides in the miRNA 5'-region) primarily with the 3′-end and more rarely with the 5′-end of messenger RNA (mRNAs) transcribed from target genes [3,4]. Their binding to mRNA leads to the repression of translation or the degradation of mRNA. The binding to target mRNAs inhibits translation or increases the degradation of mRNA in many tissues. The reduction in the amount of a specific mRNA is an important result of this molecular event. A single miRNA can target many mRNAs and influence the expression of different genes, often involved in a functionally interacting pathway [5].

In mammalian cells, miRNAs bind to the 3′-UTR of target mRNAs and induce mRNA deadenylation and decapping. MiRNAs can also interact with different regions, such as the 5′-UTR, gene promoter, and coding sequence. Pairing with the 3′-UTR appears to mediate post transcriptional silencing more effectively than pairing with the 5′-UTR or coding sequence, whereas miRNA interaction with promoter regions induces transcription [3].

In recent years, the potential of miRNAs as biomarkers for many diseases, including infectious diseases, has been widely explored [6]. Vector-borne diseases account for more than 17% of all infectious diseases, causing over 700,000 deaths a year. Mosquitoes are responsible for a large number of transmissions, with the genera Anopheles, Aedes, and Culex being the most problematic vectors. These vectors are responsible for diseases such as malaria, dengue fever, chikungunya, and Zika, among others [7,8]. Mono-infections by *Plasmodium falciparum* or arboviruses can have serious consequences for human health. Cases of simultaneous infection by malaria parasite and arboviruses are increasing, with potentially fatal consequences due to similar symptoms [9]. Understanding the regulation of post-transcriptional gene expression in coinfection can provide insights into their pathophysiology.

Studies suggest that proteins and miRNAs from mosquito saliva injected during blood sampling contribute to interactions between the vector, host, and pathogen [10]. Altered miRNA expression levels in many human diseases indicate their involvement in pathogenesis [11]. A systematic review is needed to evaluate the expression profile of miRNAs in cases of coinfection with malaria and arboviruses like Dengue virus (DENV), Chikungunya virus (CHIKV), and Zika virus (ZIKV).

This review aims to describe the current knowledge on miRNA profiles and expression in malaria and arbovirus coinfections, including viruses such as DENV, CHIKV, and ZIKV. It also aims to highlight the role of these miRNAs as potential biomarkers for pathologies associated with these infectious diseases. Understanding gene regulation in the pathogenesis of post-transcriptional gene expression pathways in these coinfections can facilitate biological validation of the activities of these medicinal plants and their miRNAs in regulating these genes.

## Materials and methods

### Study registration

This systematic review protocol has been written following the recommendation of the Preferred Reporting Items for Systematic Reviews and Meta-Analysis Protocol (PRISMA-P) 2015 guideline [12] (S1 File). It is registered in the international prospective register of systematic reviews PROSPERO system https://www.crd.york.ac.uk/PROSPERO/view/CRD42024600379. The systematic review itself will be reported following PRISMA statements, and the Cochrane Handbook for Systematic Reviews of Interventions will be used as best-practice guidance [13].

**Inclusion criteria.**

1. **Type of studies**. Randomised and non-randomised trials, case-control studies, cohort studies, and cross-sectional studies, with data on the profile of miRNAs expressed in malaria and arboviruses (DENV, CHIKV, ZIKV) coinfections, will be included. Studies assessing miRNAs in mono-infection will be used for comparison.

2. **Type of participants**. This systematic review will include studies involving patients coinfected with malaria and one or more arboviruses of interest, with no restrictions on age, gender, or location. The following cases could be grouped as coinfections, including malaria and DENV, malaria and CHIKV, malaria and ZIKV, or malaria and multiple arboviruses. Studies involving blood, plasma, serum, or other biological samples from these patients will be considered. In vitro or animal models will also be included if they explore profiles of relevant miRNAs in the context of that coinfection.

   For a secondary comparative, studies involving mono-infections (malaria alone, DENV alone, CHIKV alone, ZIKV alone) will also be included. The purpose of this is to help distinguish miRNAs that are specifically associated with the co-infection state from those that are dysregulated in response to a single pathogen. This comparative approach will aid in identifying disease-specific miRNA signatures and may help to explain sources of heterogeneity in the primary co-infection analysis.

3. **Interventions**. The evaluation of the miRNA profile will constitute the intervention. The selection of studies will be focused on the exposure to malaria in the context of coinfection with arboviruses, to characterize the relationship between the presence of a coinfection, the severity and etiology, and levels of miRNA variation.

4. **Comparators.** The study will compare the miRNA profiles of patients with coinfections to those with mono-infections and those without infection. Comparison may also involve different infection stages or disease severities, if these are reported.

5. **Types of outcomes.** The primary outcome will be to understand the different expression profile of miRNAs expressed in coinfected individuals compared to mono-infected and uninfected controls. This will also include the potential identification of miRNAs involved in the coinfection status, infection stage, or severity, and potential miRNAs relevant to diagnostic, prognostic, or therapeutic targets.

   The secondary outcome will include predicted miRNA-mRNA interactions, functional and pathway analyses of regulated biological processes, as well as the roles of deregulated miRNAs in host-pathogen interactions, particularly in immune evasion, inflammation, and disease progression.

6. **Report characteristics.** The inclusion criteria for the report characteristics encompass all studies conducted in this field from 2000 (period of the emergence of malaria-arbovirus coinfection) to the present, with no restriction on language.

   **Exclusion criteria.** The reviews, letters, comments, editorials, case series (with fewer than ten participants), preprints, and opinion papers will be excluded. Furthermore, studies not about malaria, arbovirus, or their coinfection, and those not reporting the profiling of miRNAs and lacking a clear comparator as a control group, will not be considered in the present study.

### Search strategy

No similar topics currently registered for systematic review were identified within the PROSPERO platform. Moreover, an initial search was performed using PubMed without any prior or ongoing similar systematic review published.

   The data sources will include institutional websites, grey literature, preprint servers (medRxiv, bioRxiv), and contacting corresponding authors. The search for relevant literature will encompass several electronic databases, including PubMed/MEDLINE, Web of Science, CENTRAL, Embase through the Ovid MEDLINE All, CINAHL, and Scopus. Data search strategy on some of these databases is available in the supplementary file S2 File.

   Grey literature, including conference papers, technical reports, theses, and dissertations, will be identified via Google Scholar, Google, and OpenGrey. To capture the most recent and unpublished studies, a dedicated search will also be performed in the medRxiv and bioRxiv preprint repositories via their respective websites. All databases will be searched from 2000 onwards to the present, and key terms and Medical Subject Headings (MeSH) will be specifically designed for each database. To ensure comprehensive coverage, the reference lists of existing systematic reviews will also be manually screened.

   The electronic search will explore the combinations of the keywords covering the PICO (patients, intervention, comparison, outcome) framework. The search will be conducted without language restriction, and will combine keywords, MeSH terms, and their synonyms, and all the search components will be combined with the Boolean operator "AND" while the keywords within each component will be combined with "OR". Searches will be updated immediately before the final synthesis to incorporate the most recent studies eligible for inclusion.

### Data management

All records from the various sources included in our search strategy will be combined, uploaded into the reference management software Mendeley 1.19.8, and de-duplicated. Meanwhile, titles as well as other paper information will be saved in an Excel file.

 

**Selection of studies for inclusion in the review**

Two reviewers (SM and PF) will perform duplicate and independent double extraction of the data using SysRev https://sysrev.com/, the systematic review production software used to filter the titles and abstracts initially identified. Any disagreement will be solved by either a third reviewer (ALDW) or by discussion and consensus. Two review authors (AT, ZM) will independently evaluate the full text of the selected records. Discrepancies will be resolved by consensus or by an arbitration of a third review author (TCF). The agreement between the two first review authors will be estimated by Cohen's kappa coefficient [14].

**Risk of bias assessment**

The risk of bias of included studies will be assessed using the ROBINS-I tool [15,16], which is designed to evaluate the risk of bias in non-randomized intervention studies. For randomized controlled trials, ROBIS (RoB 2.0) will be used [17] and for observational studies, the NIH quality assessment tool for observational cohort and cross-sectional studies. The Newcastle-Ottawa Scale (NOS) will be used for the cohort study [18]. Any disagreements between the review authors regarding the risk of bias assessment will be resolved through discussion and consensus or arbitration by a third author. Publication bias will be assessed visually by inspecting funnel chart asymmetry and using Egger's test. A p-value of less than 0.1 will indicate potential bias [19].

To minimize bias when selecting and extracting data from the included studies, a second pair of reviewers (AN, ENJS) will validate the electronic search by conducting an independent PubMed search using the same strategy. These authors will also review all articles excluded by the first two assessors. Any disagreements between the reviewers during the selection, data extraction, or risk of bias assessment processes will be resolved by consensus or, if necessary, by a third reviewer (ACK).

To minimize publication bias, the review will incorporate both published and unpublished data from multiple sources. When analysis includes more than ten studies, publication bias will be assessed through the visual inspection of a funnel plot. However, a specific search of clinical trial registries for unpublished diagnostic accuracy will not be conducted.

**Data extraction and management**

A standardized data extraction form will be developed and piloted to ensure consistency between the data extractors. The following data will be extracted from the included studies:

- Study characteristics: author, year of publication, country, aim of the study, study design, and clinical/study setting.

- Population characteristics: sample size, inclusion/exclusion criteria, and patient age, as well as the type of coinfection.

- Laboratory testing: index testing method, type of sample (whole blood, serum, plasma, human cell type), control group characteristics, and units of measurement.

- MiRNAs profile details: source of biological samples (whole blood, plasma, serum, cells, animals), miRNAs isolation and detection method, and a list of expressed miRNAs.

- Primary outcomes: expression level of miRNAs (upregulated or downregulated) across study groups, miRNAs associated with coinfection status, infection stage, infection severity, or diagnostic, prognostic, therapeutic potential.

- Secondary outcomes: predicted and validated miRNA-mRNA interactions, functional pathway, miRNAs involved in immune evasion, inflammation, and disease progression.

The extracted data will be cleaned and managed in Microsoft Excel. Any transformations or conversions necessary for further analysis will also be carried out here. This will include normalising expression measures, harmonizing gene and

miRNA nomenclature, and consolidating biological pathways. All data management procedures will be documented to ensure transparency and reproducibility.

## Statistical analyses and evidence synthesis

An overview of the included studies will be summarised using tables and diagrams. The findings from eligible studies will be described in a structured narrative format focusing on patterns, trends, and study characteristics. Where quantitative data are reported, descriptive statistics such as frequencies, percentages, and ranges will be presented.

For missing or unclear information, the authors will be contacted on a case-by-case basis to obtain clarifications. The degree of heterogeneity across studies will be assessed qualitatively, considering study characteristics and the consistency of findings, and categorized as none, low, moderate, or high [20,21].

To explore the potential sources of heterogeneity and test the robustness of the findings, we plan to conduct subgroup analyses based on the following factors, if enough studies are available:

*Infection type:* comparing different coinfection pairs (Plasmodium and DENV, Plasmodium and CHIKV, Plasmodium and ZIKV) and contrasting them with mono-infection profile.

*Study design:* stratifying results by observation design (cohort versus case control).

*Detection method:* comparing studies that used different miRNA profiling platforms (RNA sequencing versus microarrays versus RT-PCR).

*Sample type:* analyzing findings separately for different biological samples (whole blood versus plasma versus serum versus cells).

*Risk of bias:* A sensitivity analysis will be performed by excluding studies judged to have a high overall risk of bias to determine its influence on the synthesis.

Confidence in the evidence will be appraised using the GRADE (Grade of Recommendations, Assessment, Development and Evaluation) approach, which considers study design, risk of bias, inconsistency, indirectness, imprecision, and publication bias. Based on these domains, the certainty of the evidence will be categorized as "high", "moderate", "low", and "very low" (S3 File) [22,23]. Statistical pooling or meta-analysis will not be conducted; instead, results will be synthesized narratively, highlighting patterns, similarities, and differences across studies.

## Ethic statements and dissemination

As this study is a systematic review of previously published literature, it does not involve direct human or animal subjects. Therefore, ethical approval from an Institutional Review Board is not required. The findings of this systematic review will be disseminated through publication in a peer-reviewed scientific journal.

## Study status and timeline

The systematic review is currently ongoing.

- Record Screening: The electronic database searches are planned. The screening of full-text records, abstracts, and titles will be finished by December 2025.

- Data Extraction: As soon as the full-text screening stage is finished, data extraction from the included studies will start. It is projected that this process will be completed by February 2026.

- Results Expectation: Within eight months of the current date, data analysis, manuscript preparation, and submission to a peer-reviewed journal are expected to be finished, with results expected by May 2026.

## Discussion

Understanding the molecular regulators, such as miRNAs, that underpin the host response to such coinfections is crucial because of the overlapping epidemiology and clinical presentation of mosquito-borne diseases like dengue, chikungunya, Zika, and malaria, as well as their unique and intricate pathogenic mechanisms. By focusing on the function of miRNAs, this protocol outlines a strategy for a systematic review aimed at closing a significant gap in our molecular understanding of malaria-arbovirus coinfections. The approach outlined here is intended to be thorough and reliable. To minimize publication bias, the search strategy utilizes a range of databases, grey literature sources, and preprint servers. The incorporation of diverse observational studies ensures the inclusion of the best available evidence, while acknowledging the moral and practical difficulties of conducting trials on co-infected populations. The application of standardized instruments for evidence quality and bias risk, as well as dual, independent review at every stage.

The inherent heterogeneity across studies, which is likely to result from differences in study populations, pathogen strains, and methodology, is a major anticipated challenge. The results of miRNA quantification can be greatly impacted and synthesis made more difficult by variations in patient demographics, Plasmodium and arbovirus genotypes, sample types, miRNA isolation kits, profiling platforms, and inconsistent data reporting. Furthermore, to address the challenge of heterogeneity and to better interpret the miRNA profiles identified in co-infections, our protocol includes a planned secondary analysis of studies on mono-infections. By directly comparing the miRNA expression in co-infected individuals to those with single infections, we aim to disentangle pathogen-specific effects from those unique to the co-infection state. This approach is expected to provide a more nuanced understanding and help prioritize miRNAs that are most likely to be biomarkers or mediators of the co-infection pathophysiology.

A meta-analysis is unlikely to be suitable because of this anticipated heterogeneity. To find recurring themes and significant differences in the literature, a narrative synthesis will be carried out, classifying results by virus type, sample matrix, and disease severity. A comprehensive and nuanced map of the current situation will be produced by this method.

In addition to providing an overview of known human miRNAs, this review will actively explore a novel idea: the possibility that plant-derived miRNAs could regulate genes across kingdoms. Medicinal plant consumption is common among populations in endemic areas, and these exogenous miRNAs may affect host gene expression. Examining this potential could reveal a completely new mechanism of host-pathogen-plant interaction and pave the way for the creation of ground-breaking miRNA-based treatments made from natural products.

Furthermore, the miRNA signatures that will be cataloged in this review are expected to provide a foundation for developing novel therapeutic strategies. Dysregulated miRNAs identified in co-infected or monoinfected individuals will represent potential drug targets; downregulated miRNAs could be restored using synthetic miRNA mimics, while overexpressed miRNAs could be suppressed with anti-miRNA oligonucleotides (antagomirs) [24]. A significant translational challenge for such RNA-based therapies will be achieving efficient and targeted delivery [25]. Various nanocarrier platforms have been explored for this purpose. These include biocompatible polymers (chitosan), which offer mucoadhesive properties and controlled release; inorganic nanoparticles (carbon quantum dots, layered double hydroxides), known for their stability and ease of functionalization; and peptide-based systems (cell-penetrating peptides, branched amphiphilic peptide capsules), which enhance cellular uptake and endosomal escape [26,27]. Alongside these, lipid-based nanoparticle systems have been extensively developed. Among these, lipid nanoparticle systems, have emerged as a leading solution, offering protection against degradation, enhanced cellular uptake, and the potential for tissue-specific targeting, as supported by preclinical studies in other infectious and inflammatory conditions [28,29]. Therefore, cataloging dysregulated miRNAs in malaria-arbovirus co-infections will constitute a critical first step toward future research exploring these miRNAs as therapeutic targets, potentially delivered via advanced nanocarriers.

Thus, this systematic review will be the first to synthesize data regarding the miRNA landscape in coinfections between malaria and arbovirus. Additionally, it will directly inform future hypothesis-driven studies that aim to validate miRNAs as

therapeutic targets or diagnostic biomarkers, in addition to acting as a foundational resource for researchers. It lays the groundwork for important contributions by providing a rigorous methodology to investigate both endogenous and exogenous miRNA involvement.

## Limitations of the review process

This systematic review is expected to have several limitations. The scope and reliability of our findings may be constrained by the small number of relevant studies that the emerging field of miRNA research in coinfections may produce. A narrative synthesis is required because of the anticipated significant heterogeneity in clinical and methodological aspects, such as patient demographics, sampling strategies, and laboratory techniques, which will hinder quantitative meta-analysis. Despite being suitable for mapping evidence, this method might make it more difficult to reconcile differences between studies. Despite thorough searches of the grey literature, publication bias may still exist because studies with null results are frequently underrepresented. Egger's test and funnel plots will be used to assess this risk if there are enough studies available. Lastly, the reporting quality of the included studies determines how accurate the data extraction and synthesis are; gaps may be introduced by missing or unclear data.

## Conclusion

In summary, this protocol outlines a comprehensive and methodologically rigorous plan for the first systematic review aimed at mapping the landscape of miRNA expression in malaria and arbovirus co-infections. By synthesizing available data from both co-infected and mono-infected populations and proactively exploring sources of heterogeneity through subgroup analyses, this review will enable the identification of key miRNA signatures associated with these complex immunological interactions. The findings should support the development of new hypotheses regarding the pathogenesis of co-infections, highlight potential diagnostic and prognostic biomarkers, and suggest novel therapeutic targets. Furthermore, the miRNA profiles highlighted in this review will guide the future development of RNA-based therapies, especially those utilizing a nanoparticle delivery system to improve stability, specificity, and translational potential. Ultimately, this work will constitute a fundamental resource essential for guiding future research aimed at improving the management and outcomes of patients with these overlapping infectious diseases.

## Supporting information

**S1 File. PRISMA-P checklist.** Checklist showing the location of Preferred Reporting Items for Systematic Reviews and Meta-Analysis Protocol (PRISMA-P) items in the manuscript.
(DOCX)

**S2 File. Search strategy.** Detailed search strategy for one major database (PubMed, Scopus, Web of Science, CENTRAL), which will be adapted for other databases.
(DOCX)

**S3 File. GRADE summary.** Template for summarizing the assessment of the certainty of evidence using the GRADE approach.
(DOCX)

## Acknowledgments

The authors would like to thank the Malaria Research Capacity Development Consortium (MARCAD) and the Biotechnology Centre of the University of Yaoundé for hosting the work sessions of drafting the systematic review protocol, and all the authors for the work already done in this field.

## Author contributions

**Conceptualization:** Andrillene Laure Deutou Wondeu, Stefano Pirrò.

**Supervision:** Andrea Galgani, Vittorio Colizzi, Wilfred Fon Mbacham.

**Writing – original draft:** Andrillene Laure Deutou Wondeu, Alex Durant Nka, Aude Christelle Ka'e, Ezechiel Ngoufack Jagni Semengue.

**Writing – review & editing:** Andrillene Laure Deutou Wondeu, Alex Durant Nka, Aude Christelle Ka'e, Ezechiel Ngoufack Jagni Semengue, Sylvanie Masso, Pascal Fisemou, Adegoke Taiwo Mobolaji, Zeekah Marymag, Calvino Fomboh Tah, Rhoda Nsen Bughe, Akindeh Mbu Nji, Andrea Galgani, Stefano Pirrò, Vittorio Colizzi, Wilfred Fon Mbacham.

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
