## [Decision Letter · Decision Letter 0]

12 Nov 2025

Dear Dr. Andrillene Laure Deutou Wondeu,

Thank you for submitting your manuscript to PLOS ONE. After careful consideration, we feel that it has merit but does not fully meet PLOS ONE’s publication criteria as it currently stands. Therefore, we invite you to submit a revised version of the manuscript that addresses the points raised during the review process.

We look forward to receiving your revised manuscript.

Kind regards,

Jisheng Liu, Ph.D.

Academic Editor

PLOS ONE

Journal Requirements:

Additional Editor Comments:

This manuscript describes a systematic review protocol that will characterize the profiles of microRNAs involved in malaria and arbovirus coinfections, which will advance understanding of coinfection pathogenesis. Besides miRNAs, siRNAs or dsRNAs also regulate gene expression, especially the first nanoparticle encapsulated dsRNA was reported in mosquitoes. Therefore, could nanoparticles also be employed to encapsulate miRNAs as well? Maybe the recent review (Recent progress in nanoparticle-mediated RNA interference in insects: Unveiling new frontiers in pest control) could be used to discuss this possibility.

Reviewers' comments:

Reviewer's Responses to Questions

**Comments to the Author**

1. Does the manuscript provide a valid rationale for the proposed study, with clearly identified and justified research questions?

Reviewer #1: Yes

Reviewer #2: Yes

2. Is the protocol technically sound and planned in a manner that will lead to a meaningful outcome and allow testing the stated hypotheses?

Reviewer #1: Yes

Reviewer #2: Yes

3. Is the methodology feasible and described in sufficient detail to allow the work to be replicable?

Reviewer #1: Yes

Reviewer #2: Yes

4. Have the authors described where all data underlying the findings will be made available when the study is complete?

Reviewer #1: Yes

Reviewer #2: Yes

5. Is the manuscript presented in an intelligible fashion and written in standard English?

Reviewer #1: Yes

Reviewer #2: Yes

Reviewer #1: I suggest this change in line 69-70: "Pairing with the 3′-UTR appears to mediate post

transcriptional silencing more effectively than pairing with the 5'-UTR or coding sequence,

whereas miRNA interaction with promoter regions induces transcription [3]. "

In line 85 the author cite the viruses by their abbreviations (CHIKV, ZIKV, …) but you have never introduced them before. The first time you mention them, include the full name followed by the abbreviation in parentheses. Example: Chikungunya virus (CHIKV), Zika virus (ZIKV), Dengue virus (DENV). Thereafter, you can use just the abbreviation (for instance, “CHIKV replication…”).

In the search strategy/risk of bias assessment/data extraction and management paragraphs, I suggest replacing “we did” with an impersonal construction, which is more appropriate in an academic context. For example, instead of “We did an analysis,” write “An analysis was conducted.”

A conclusion paragraph is missing.

Reviewer #2: This is an interesting article on a very important and timely topic, and it is well planned. Although I have a few suggestions for the authors to consider, which may help them strengthen the outcomes of their work:

1. In addition to studies on co-infections, consider including studies investigating single infections (e.g., malaria or dengue) as a secondary analysis for meta-analysis. This may help identify disease-specific miRNAs and reduce heterogeneity in the overall synthesis.

2. Broaden the database searches and include preprint servers (e.g., medRxiv, bioRxiv) to capture recent or unpublished studies and minimize publication bias.

3. Clearly describe how missing or ambiguous data will be addressed (e.g., by contacting study authors) to improve data reliability.

4. Plan subgroup or sensitivity analyses based on infection type, study design, or detection methods to better interpret variability across studies.

**Do you want your identity to be public for this peer review?** For information about this choice, including consent withdrawal, please see our Privacy Policy

Reviewer #1: No

Reviewer #2: No

---

## [Author Response · Author response to Decision Letter 1]

29 Nov 2025

Response to the Reviewers

Manuscript ID: PONE-D-25-47997

MicroRNAs profiling in malaria and arbovirus coinfection: A systematic review protocol.

We thank the editors and reviewers for their thoughtful comments and constructive suggestions on our manuscript. We have carefully revised the manuscript to address all points raised. The changes are highlighted in the "Revised Manuscript with Track Changes" file. Below is our point-by-point response.

Journal requirements revisions

In addition to the specific points raised by the reviewers, we have also ensured full compliance with PLOS ONE's journal formatting requirements:

-The ethics statement has been placed exclusively in the 'Ethics and dissemination' subsection within the Methods.

-Formal captions for all Supporting Information files have been added at the end of the manuscript.

-The reference list has been thoroughly checked and formatted according to the Vancouver/ICMJE style.

Academic Editor Comments

Comment: This manuscript describes a systematic review protocol that will characterize the profiles of microRNAs involved in malaria and arbovirus coinfections, which will advance understanding of coinfection pathogenesis. Besides miRNAs, siRNAs or dsRNAs also regulate gene expression, especially the first nanoparticle encapsulated dsRNA was reported in mosquitoes. Therefore, could nanoparticles also be employed to encapsulate miRNAs as well? Maybe the recent review (Recent progress in nanoparticle-mediated RNA interference in insects: Unveiling new frontiers in pest control) could be used to discuss this possibility.

Response: We thank the editor for this insightful suggestion regarding the potential of nanoparticle-based delivery systems for miRNAs. While our systematic review protocol is primarily focused on the identification and profiling of miRNA signatures, we acknowledge the therapeutic potential of this technology. In response, we have integrated a mention of this promising future application into the abstract's conclusion to highlight the broader impact of our findings. Lines 47-48

Reviewer #1 Comments

Comment 1: I suggest this change in line 69-70: 'Pairing with the 3′-UTR appears to mediate post transcriptional silencing more effectively than pairing with the 5'-UTR or coding sequence, whereas miRNA interaction with promoter regions induces transcription [3].

Response: We have rephrased the sentence as suggested for better clarity.

Changes Made: The sentence in the Introduction now reads: Pairing with the 3′-UTR appears to mediate post transcriptional silencing more effectively than pairing with the 5'-UTR or coding sequence, whereas miRNA interaction with promoter regions induces transcription [3]. Lines 74-76

Comment 2: In line 85 the author cite the viruses by their abbreviations (CHIKV, ZIKV, …) but you have never introduced them before. The first time you mention them, include the full name followed by the abbreviation in parentheses. Example: Chikungunya virus (CHIKV), Zika virus (ZIKV), Dengue virus (DENV). Thereafter, you can use just the abbreviation (for instance, “CHIKV replication…”).

Response: We have now introduced the full virus names with their abbreviations upon first mention in the abstract.

Changes Made: Added in the Abstract, Background section: "Coinfections between malaria and arboviruses such as Dengue virus (DENV), chikungunya Virus (CHIKV), and Zika virus (ZIKV)..." Lines 23-24, 91-92

Comment 3: In the search strategy/risk of bias assessment/data extraction and management paragraphs, I suggest replacing “we did” with an impersonal construction, which is more appropriate in an academic context. For example, instead of “We did an analysis,” write “An analysis was conducted.”

Response: We have revised the methods section to use impersonal constructions throughout.

Changes Made: Changed to passive voice/impersonal constructions in the Search Strategy, Risk of Bias Assessment, and Data Extraction and Management sections. Lines 150-171; 184-224

Comment 4: "A conclusion paragraph is missing."

Response: We have added a comprehensive Conclusion paragraph to the main text.

Changes Made: Added a new Conclusion section summarizing the protocol's objectives, methodology, and expected significance. Lines 325-335

Reviewer #2 Comments

Comment 1: "In addition to studies on co-infections, consider including studies investigating single infections (e.g., malaria or dengue) as a secondary analysis for meta-analysis. This may help identify disease-specific miRNAs and reduce heterogeneity in the overall synthesis."

Response: We agree that including mono-infection studies will enhance the interpretation of co-infection-specific miRNA profiles. We have explicitly included this as a key part of our methodology.

Changes Made: Added a new subsection under "Type of participants" in the Inclusion Criteria detailing the inclusion of mono-infection studies for a secondary comparative analysis. Lines 120-125; 289-296

Comment 2: "Broaden the database searches and include preprint servers (e.g., medRxiv, bioRxiv) to capture recent or unpublished studies and minimize publication bias."

Response: We have expanded our search strategy to include major preprint servers.

Changes Made: Added medRxiv and bioRxiv to the list of data sources in the Search Strategy and described the planned search process. Lines 154-155, 280

Comment 3: "Clearly describe how missing or ambiguous data will be addressed (e.g., by contacting study authors) to improve data reliability."

Response: We have clarified the procedure for handling missing or unclear data.

Changes Made: Explicitly stated in the "Statistical analyses and evidence synthesis" section that corresponding authors will be contacted on a case-by-case basis to obtain clarifications. Lines 230-231

Comment 4: "Plan subgroup or sensitivity analyses based on infection type, study design, or detection methods to better interpret variability across studies."

Response: We have now pre-specified plans for subgroup and sensitivity analyses to proactively investigate sources of heterogeneity.

Changes Made: Added a new dedicated paragraph in the "Statistical analyses and evidence synthesis" section detailing planned subgroup analyses by infection type, study design, detection method, sample type, and risk of bias. Lines 234-245

We believe these revisions have significantly strengthened the manuscript and thank the editors and reviewers for their valuable input.

---

## [Editor Report · Decision Letter 1]

2 Dec 2025

Dear Dr. Deutou Wondeu,

Thank you for submitting your manuscript to PLOS ONE. After careful consideration, we feel that it has merit but does not fully meet PLOS ONE’s publication criteria as it currently stands. Therefore, we invite you to submit a revised version of the manuscript that addresses the points raised during the review process.

We look forward to receiving your revised manuscript.

Kind regards,

Jisheng Liu, Ph.D.

Academic Editor

PLOS ONE

**Journal Requirements:**

**Additional Editor Comments:**

In abstract L47-48 "Additionally, the miRNA profiles identified may provide foundations for developing novel therapeutic approaches, including nanoparticle-based delivery systems." This claim should be indicated in the discussion with supporting citations, as well as in the conclusion paragraph.

---

## [Author Response · Author response to Decision Letter 2]

4 Dec 2025

Dear Dr. Jisheng Liu,

Thank you for the opportunity to revise our manuscript and for the constructive feedback from the editorial team. We have carefully considered the comments and have made revisions to strengthen the manuscript accordingly. Please find below our point-by-point response.

Response to Journal Requirements

1. We have reviewed the reviewer comments and confirm that no specific citation recommendations were made that require the addition of new references.

2. We have thoroughly reviewed our reference list to ensure it is complete and correct. We confirm that our manuscript contains no retracted articles. As part of our revision to address the editor's comment, we have added four new references to the Discussion section.

We have also corrected bibliographic information for reference 23, which was previously inaccurate, and updated the access dates for references 13, 16, and 22.

Response to the Academic Editor:

Comment: In abstract L47-48, "Additionally, the miRNA profiles identified may provide foundations for developing novel therapeutic approaches, including nanoparticle-based delivery systems. This claim should be indicated in the discussion with supporting citations, as well as in the conclusion paragraph.”

Response: We thank the editor for this important suggestion. We agree that the therapeutic potential of the identified miRNA profiles requires proper contextualization within the existing literature. In response:

1. We have expanded the Discussion section in the paragraph now on page 11 (lines 307-318) to explicitly address the translational potential of miRNA signatures. We now discuss how dysregulated miRNAs can serve as therapeutic targets, citing relevant examples from other fields (Citation 24-25) and outline the rationale for nanoparticle-based delivery systems to overcome key challenges in RNA therapy, such as stability and targeted delivery (Citation 26-27).

2. We have confirmed and slightly reinforced this point in the Conclusion paragraph (page 15, line 345-348) to ensure it is presented as a clear and supported implication of our proposed systematic review.

These changes provide the necessary scholarly support for the claim made in the Abstract and enhance the manuscript's discussion of future applications.

We have also performed a final check to ensure the reference list is complete and correct.

We believe these revisions have significantly improved the manuscript and hope it now meets the high publication standards of PLOS ONE.

Thank you again for your time and consideration.

Sincerely,

Deutou Wondeu

---

## [Editor Report · Decision Letter 2]

19 Dec 2025

Dear Dr. Deutou Wondeu,

Thank you for submitting your manuscript to PLOS ONE. After careful consideration, we feel that it has merit but does not fully meet PLOS ONE’s publication criteria as it currently stands. Therefore, we invite you to submit a revised version of the manuscript that addresses the points raised during the review process.

We look forward to receiving your revised manuscript.

Kind regards,

Jisheng Liu, Ph.D.

Academic Editor

PLOS One

Additional Comments:

The authors claim "Nanoparticle-based systems, particularly lipid nanoparticles have emerged as a leading solution". Some sentences regarding various nanomaterials with recent supported reference should be added before this sentence, including chitosan, star polycations, carbon quantum dots, layered double hydroxides guanylated polymers branched amphiphilic peptide capsules, cell-penetrating peptides etc. Then the author can narrow down to lipid-based nanoparticles.This will provide general information to specific for readers.

---

## [Author Response · Author response to Decision Letter 3]

22 Dec 2025

Response to Academic Editor

Comment: The authors claim "Nanoparticle-based systems, particularly lipid nanoparticles have emerged as a leading solution". Some sentences regarding various nanomaterials with recent supported reference should be added before this sentence, including chitosan, star polycations, carbon quantum dots, layered double hydroxides guanylated polymers branched amphiphilic peptide capsules, cell-penetrating peptides etc. Then the author can narrow down to lipid-based nanoparticles. This will provide general information to specific for readers.

Response: We sincerely thank the Academic Editor, Dr. Liu, for this precise and valuable suggestion, which stems from his deep expertise in the field. We fully agree that providing this technological context will benefit readers.

In direct response, we have revised the relevant sentence in the Discussion section. We now explicitly state that a broad range of nanocarrier platforms-encompassing the very categories mentioned (natural polymers, inorganic materials, peptide-based systems, etc.) have been engineered for nucleic acid delivery, as recently and comprehensively reviewed for RNA interference applications (new Reference 26 and 27). This provides the perfect foundational context to then logically focus on lipid nanoparticles as a leading therapeutic delivery platform.

We are confident that this addition, anchored by your authoritative review, fully addresses the comment and enhances the manuscript's discussion of future translational potential.

---

## [Editor Report · Decision Letter 3]

26 Dec 2025

MicroRNAs profiling in malaria and arbovirus coinfection: A systematic review protocol.

PONE-D-25-47997R3

Dear Dr. Andrillene Laure Deutou Wondeu,

We’re pleased to inform you that your manuscript has been judged scientifically suitable for publication and will be formally accepted for publication once it meets all outstanding technical requirements.

Kind regards,

Jisheng Liu, Ph.D.

Academic Editor

PLOS One

---

## [Editor Report · Acceptance letter]

PONE-D-25-47997R3

PLOS One

Dear Dr. Deutou Wondeu,

I'm pleased to inform you that your manuscript has been deemed suitable for publication in PLOS One. Congratulations! Your manuscript is now being handed over to our production team.

Kind regards,

on behalf of

Professor Jisheng Liu

Academic Editor

PLOS One